# Impact of Data Loss on Multi-Step Forecast of Traffic Flow in Urban Roads Using K-Nearest Neighbors

**Amin Mallek \*** , **Daniel Klosa and Christof Büskens**

WG Optimisation and Optimal Control, Center for Industrial Mathematics, University of Bremen,
28359 Bremen, Germany
\* Correspondence: amallek@uni-bremen.de

**Abstract:** Data-driven models have recently proved to be a very powerful tool to extract relevant information from different kinds of datasets. However, datasets are often subject to multiple anomalies, including the loss of important parts of entries. In the context of intelligent transportation, we examine in this paper the impact of data loss on the behavior of one of the frequently used approaches to address this kind of problems in the literature, namely, the k-nearest neighbors model. The method designed herein is set to perform multi-step traffic flow forecasts in urban roads. In our study, we deploy non-prepossessed real data recorded by seven inductive loop detectors and delivered by the Traffic Management Center (VMZ) of Bremen (Germany). Firstly, we measure the performance of the model on a complete dataset of 11 weeks. The same dataset is then used to artificially create 50 incomplete datasets with different gap sizes and completeness levels. Afterwards, in order to reconstruct these datasets, we propose three computationally-low techniques, which proved through empirical testing to be efficient in reproducing missing entries. Thereafter, the performance of the E-KNN model is assessed under the original dataset, incomplete and filled-in datasets. Although the accuracy of E-KNN under incomplete and reconstructed datasets depends on gap lengths and completeness levels, under original dataset, the model proves to deliver six-step forecasts with an accuracy of 83% on average over 3 weeks of the test set, which also translates to a less than one car per minute error.

**Keywords:** data loss; incomplete dataset; intelligent transportation; k-nearest neighbors; linear regression; short-term forecast; traffic flow



## 1. Introduction

A huge part of business and economy nowadays heavily relies on transportation systems. Among others is e-commerce, which is partly based on delivering goods to customers, and transportation of individuals from and to work, for example. Optimizing costs and time for such operations requires efficient and intelligent transportation management systems. One of the crucial components of these systems is the prediction of different attributes related to traffic, especially traffic flow volume. Often, this latter is the main element on which other traffic-related features are based, and is a vital topic discussed in both academia and industry. Traffic managers usually depend on short-term traffic flow forecasts to plan and formulate efficient strategies in order to alleviate road congestion and further optimize vehicular traffic inside cities. Moreover, travelers also refer to these forecasts to take decisions about their traveling plans. The development of approaches for the purpose of accurate short-term flow forecasting might not be successful without a large amount of data. Therefore, traffic management centers deploy a large range of tools to monitor and record traffic attributes, including inductive loop detectors, video and image processing, radars of different kinds, and other Internet of Things (IoT) mechanisms. Recently, and in the past years, researchers have extensively examined the problem of short-term traffic flow forecasts. Consequently, several data-driven models have been proposed to tackle

this problem, categorized into two main classes: parametric and non-parametric models. We can briefly describe parametric models as the models that output predictions based on an explicit function defined within a finite set of parameters. These parameters are often estimated by training the model with a given dataset, for instance: ARIMA and its variants [1–3], neural networks [4–6], deep learning [7], and linear regression [8]. In contrast, non-parametric methods deliver predictions without assuming any prior knowledge or having any explicit formulas, such as support vector regression (SVR) [9,10] and k-nearest neighbors [11–13].

In the literature, researchers generally consider datasets with a few missing entries that are usually imputed using simple techniques, otherwise, they take into account only valid data. However, only a few of them drew attention to the impact of data loss on the performance of predictive models. Therefore, some techniques have been used in the literature to substitute corrupted and missing entries with valid data. First attempts were made, for instance, by Nihan et al. [14] by deploying the classical auto-regressive integrated moving average model. Zhong et al. [15] proposed different techniques to substitute missing input, including neural networks and regression models. Later on, a non-parametric spatio-temporal kernel regression model is developed to forecast travel time under the assumption of sensor malfunction. The results were compared to a k-nearest neighbors model, which is also non-parametric. The k-nearest neighbors technique has also been used for traffic data imputation in [16]. Tian et al. proposed a long short-term memory-based neural network that efficiently circumvents the negative impact of data loss [17]. Duan et al. [18] employed a deep learning-based approach called denoising stacked autoencoders for efficient imputation of missing data. Teresa Pamula [19] investigated the sensitivity of neural networks to loss of data in traffic flow prediction and proposed a strategy to substitute lost data in a way where the accuracy of forecasts is maintained. Some statistical models, including Markov chains, PPCA-based approaches, and Monte Carlo simulations have also been used [20–23]. An automated imputation procedure based on an adaptive identification technique that tries to minimize the error between simulated and measured densities was elaborated by Muralidharan and Horowitz [24]. Other techniques such as replacement by null values, substituting by the sample mean, or exponentially moving average were considered for testing and they showed good performance practically [25,26]. Several other strategies have as well been used including fuzzy C-means hybridized with a genetic algorithm [27], tensor-based methods [28], and simulator software such as Sumo and TransWorld [29].

In this paper, we address the problem of multi-step flow volume forecast in urban roads under the circumstances of data loss. This problem is part of the DiSCO2 project conducted at the University of Bremen. The aim of this project is to set the stage for decision makers to take actions to reduce $CO_2$ emissions in Bremen (Germany). To this end, we develop an enhanced k-nearest neighbors model based on traffic features. This method (shortened as KNN), is known to be a non-parametric data-driven model and has extensively been investigated in the literature. Old KNN models mainly focus on single-step forecasts [30–32]. However, this technique and other non-parametric models have the advantage of being flexible and easily extensible. Therefore, herein we extend the classical k-nearest neighbors to what we call enhanced KNN, referred to as the E-KNN model. This latter takes into account more attributes related to traffic to improve forecasting accuracy. Several improvements to the KNN model have already been considered in the context of traffic flow in some papers, including [12], where the authors deployed a weighted Gaussian method to compute forecasts instead of the typical ones, as in [31]. The authors in [11] incorporated a time constraint in neighbor selection and a minima distance to avoid the selection of highly auto-correlated candidates. Cheng et al. [13] developed a KNN model based on the assumption that traffic between adjacent road segments within assigned time periods is not correlated. This spatio-temporal approach comprehensively considers the spatial heterogeneity of traffic. Our E-KNN takes into account a search radius to ensure that selected profiles share similar characteristics. It also assumes that the flow is not only distinct between weekends and working days, but also among all

weekdays. Usually, studies are carried out on processed, filtered, or normalized data, herein we measure the performance of our designed technique using raw data, provided by the Traffic Management Center (VMZ) of Bremen, with no preprocessing, meaning that noise, corrupted data, and outliers are kept as they are in our dataset. The purpose behind this is to have an idea about the accuracy of the model when it is operated online, as needed in our project. Furthermore, for the same reason, the model performs six-step (1 h) forecasts at once in order to reduce the computational time.

In the second part of the paper, we take out the same dataset used to measure the performance of E-KNN and create artificially incomplete datasets. To do so, we try to simulate the actual status of most raw datasets (including ours). Thus, we produce 50 datasets with different gap sizes and completeness levels. Afterward, we try to reconstruct the missing parts of these datasets by deploying three different techniques that we designed for this purpose. We first assess the accuracy of reconstructing these datasets and profoundly examine their structure, then apply E-KNN to each of them. At this point, we can obtain an overview of how the E-KNN model behaves when is applied to incomplete and partially reconstructed datasets. A deep analysis of this latter is thoroughly reported afterward.

The rest of this paper is structured as follows. In Section 2 we describe the basic framework of k-nearest neighbors, then introduce the enhanced version of this model, referred to as E-KNN. Section 3 comprises the imputation techniques designed to fill in incomplete datasets. An in-depth description of the dataset used in this paper, and further in some parts of our project, is sketched in Section 4. The way the incomplete datasets are created and reconstructed is also extensively reported in the same section. Afterward, we detail in Section 5 the empirical findings out of testing the performance accuracy of E-KNN under original as well as incomplete and filled-in datasets. Finally, the paper is concluded in Section 6.

## 2. K-Nearest Neighbors Model

The k-nearest neighbors model is one of the famous data-driven predictive models applied to different problems present in the literature. In its essence, this approach explores historical data to fetch patterns in the past similar to a present one. The algorithm then tries to generate forecasts based on the future states of the historical patterns that are hypothetically closely similar to a current state. The quality of this model as any other data-driven model depends on how big the data is and also how well it is represented. In what follows, we detail the basic components of KNN and how they are characterized. Afterward, based on the nature of our problem, we integrate into the model some enhancements to improve its performance.

### 2.1. Basic KNN and Notations

In this subsection, we introduce the set of notations used to describe our KNN model and its functionalities. First of all, traffic flow volume at instant $t$ at some detector (sensor) $d$ is denoted by $f_d(t)$. Thus, a flow volume series at some detector $d$ in a given time frame between $t_i$ and $t_j$ $(i \leq j)$ is defined by the following vector:

$$v_d(t_i, t_j) = [f_d(t_i), f_d(t_{i+1}), \ldots, f_d(t_{j-1}), f_d(t_j)] \tag{1}$$

For simplicity, we concisely write $f(t)$ and $v(t_i, t_j)$, unless the detector is required to be mentioned. We define a state vector of length $l$ at instant $t$ as a vector of flow volume comprising the traffic flow from instant $t$ backward to instant $t - l$, therefore it can be seen as:

$$v(t - l, t) = [f(t - l), f(t - l + 1), \ldots, f(t - 1), f(t)] \tag{2}$$

The forecasts of the next $s$ steps are given in a prediction vector denoted by $v'$ and expressed as follows:

$$v'(t + 1, t + s) = [f'(t + 1), f'(t + 2), \ldots, f'(t + s - 1), f'(t + s)] \tag{3}$$

Once the state vector is defined, the other components of the KNN framework can then be set. The first task is to select a set of nearest neighbors to a given state vector based on measuring a certain distance between this latter and other candidate vectors. The best $k$ candidates (neighbors) are then selected to be considered in the prediction process. Various methods are usually used to determine the closeness of the current state vector to other vectors or to produce forecasts. The distance between two state vectors is commonly given by the Euclidean distance, however sometimes and in some cases, the correlation coefficient distance is also considered where its superiority is proven [11]. Averaging the $k$ nearest neighbors is often the common way used to predict future states. Though, many other approaches were also applied in the literature to obtain forecasts, including weighting the $k$ nearest neighbors according to their distance to current state vector [33] and Gaussian-weight distance [12,13].

Our KNN algorithm is designed to perform multi-step prediction, therefore, at some instant $t$, the vector aggregating current state, and $s$ future steps can be seen as follows:

$$E(t) = v(t-l,t) + v'(t+1,t+s) = [f(t-l),\ldots,f(t),f'(t+1),\ldots,f'(t+s)] \quad (4)$$

In our study, we use the Euclidean distance to rank the neighbors of state vectors. Firstly, the correlation coefficient distance proved to be inferior to the Euclidean distance through the experiments. Secondly, the Gaussian-based distance did not increase the accuracy of the predictions. We also use simple averaging of the $k$ nearest neighbors. Note that weighting of neighbors procedure has also been tested and gave the same results as simple averaging. Mathematically, the Euclidean distance, in our case, is given as follows:

$$dist^{(i,j)}(v^i,v^j) = \sum_{\lambda=0}^{l} (f^i(t-\lambda) - f^j(t-\lambda))^2 \quad (5)$$

Such that $v^i$ and $v^j$ are two state vectors. As said before, our KNN is designed to forecast multiple steps, hence the formula to deliver a prediction vector is:

$$v'(t+1,t+s) = [f'(t+1),\ldots,f'(t+s)] = [\sum_{i=1}^{k} f^{(i)}(t+1)/k,\ldots,\sum_{i=1}^{k} f^{(i)}(t+s)/k] \quad (6)$$

### 2.2. Enhanced KNN

The classical KNN model has extensively been applied to different kinds of problems other than traffic flow prediction. The KNN framework commonly used is the one just described. However, several improvements can be introduced based on the problem's nature. In our problem, various features can be regarded in order to boost forecast accuracy. Therefore to enhance the performance of our KNN model, henceforth referred to as E-KNN, we incorporate the following characteristics as well:

- Detector-wise: although the flow differs from one detector to another as shown in Figure 1, it may happen that patterns from different detectors have a partial similarity. Since these candidates are retrieved from different detectors, the other parts of them may be very different, which badly impacts the forecasting accuracy. Thus, if the prediction is to be made for a given detector, the model inspects only data related to it.
- Weekday-wise: when our model explores the historical data to retrieve state vector profiles, it only considers the same weekday. For instance, if the current state vector is taken from a Wednesday, all the profiles are constructed from historical data belonging to Wednesdays. Observations showed that there is a clear difference between working-day and weekend flows. More precisely, even days of the same category differ in flow patterns, which justifies our choice. This weekday-wise pre-selection of state vector profiles showed a significant improvement in the model's performance through preliminary experiments.

- State vector length: the length of state vectors, denoted by $l$, indicates how far backward from a given instant $t$ the data is relevant to make accurate predictions. Hence, a state vector of length $l$ is given by:

$$v(t-l,t) = [f(t-l), f(t-l+1), \ldots, f(t-1), f(t)] \tag{7}$$

  The length of state vectors impacts the prediction quality as well. If $l$ is relatively small, the information provided by the state vector may be insufficient to make accurate predictions. However, a longer state vector might also provide irrelevant information. To choose the best value for $l$, preliminary experiments have been launched with different values of $l$, such that $l \in \{20, 30, 40, 50, 60, 90, 120, 150\}$. Tests showed that $l = 60$ min (six timestamps) is the best choice for our dataset.

- Search radius: to ensure that state vector profiles share similar characteristics with the current state vector, we only consider profiles within a certain radius denoted by $R$. This means that the model selects profiles falling no further than $r$ timestamps forwards and backwards from a current instant $t$. Therefore, the search space is constrained within $t - r$ and $t + r$. Obviously, as we decrease $R$, the ratio of profiles closer to the current state vector in terms of characteristics increases, and vice versa. One issue can be raised here, when $R$ becomes smaller, profiles become fewer, which may also affect the prediction accuracy. Consequently, a trade-off value of $R$ has to be determined in this respect. Experiments included $R \in \{40, 50, 60, 90, 120, 150, 200, 300\}$ and showed that $R = 90$ min ($r = 9$ timestamps) is the best search radius for our experiments. Note that above 200 min (20 timestamps), the efficiency drastically decreases, which indicates that search radius imposition is worthy.

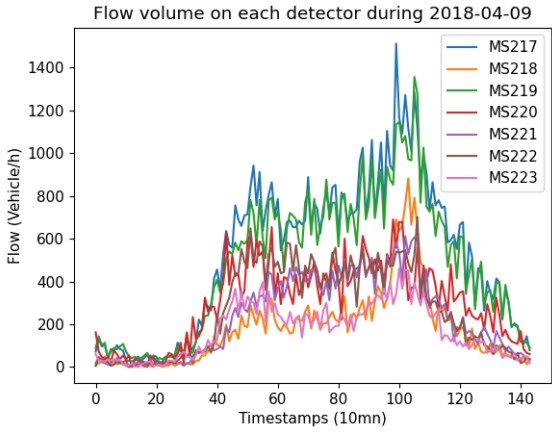

**Figure 1.** Flow volume on detectors MS217–MS223 on 9 April 2018.

## 3. Imputation Techniques

Any kind of time series generally contains a mix of invalid values, missing entries, and outliers; flow time series are not an exception. Commonly, missing and corrupted values are replaced by some kind of constant depending on the predictive model to be used, however, usually outliers are replaced with the closest rational value. In this section we introduce three different techniques with a low computational effort to be deployed for imputing missing values in our time series. The most efficient strategy among them will be used to impute missing entries in our project. The reason behind seeking low computational effort is that the chosen method is to be integrated into an online system.

### 3.1. Mean

An intuitive approach is to compute the mean of each timestamp over the entire dataset. This roughly gives the amount of flow at each period of time during the day. Without loss of generality, for 10 min accumulation we have 144 timestamps per day, we denote their flow value by $f(t)$, $t \in \{1, 2, \ldots, 144\}$. Let us assume that in our dataset we

have $D$ days and only $\omega$ available values of $f(t)$ (the rest is missing). Thus, we replace the missing flow values at timestamp $t$ in our dataset by:

$$m(t) = \frac{\sum_{i=1}^{\omega} f(t)^{(i)}}{\omega} \tag{8}$$

Such that $f(t)^{(i)}$ is the $i^{th}$ day with available flow at this timestamp ($i^{th}$ available flow value at timestamp $t$). Note that this technique excludes any differences between weekday flows.

*3.2. Mean per Weekday*

This method is a more precise approach than the previous one. Herein, besides the mean, we take into account weekdays as well. This means we compute the mean for each timestamp of each weekday. If we take for instance 10 min accumulation, then we have to take into account $144 \times 7$ values. As mentioned above the flow differs from one weekday to another, especially during weekends. The idea is similar to the previous one, over the dataset, we compute the mean flow for each timestamp related to weekday $j \in \{1, \ldots, 7\}$. Consequently, the values can be given by the following formula:

$$m^{(j)}(t) = \frac{\sum_{i=1}^{\omega(j)} f(t)^{(i)}}{\omega(j)} \tag{9}$$

where $\omega(j)$ is the number of available values over the dataset at timestamp $t$ of weekday $j$ and $m^{(j)}(t)$ is the imputed value at timestamp $t$ of weekday $j$.

*3.3. Linear Regression*

In this subsection, we briefly describe a regression analysis model that has been introduced for flow prediction in [8]. A linear regression model has been deployed, in which forecasts are given by a set of polynomials of different degrees. The general formula of these polynomials is:

$$P(x) = \sum_{k=0}^{n} \alpha_k x^k + \epsilon = \alpha_0 + \alpha_1 x + \alpha_2 x^2 + \ldots + \alpha_n x^n + \epsilon \tag{10}$$

where $n$ is the degree of $P$ and $\alpha$ is a vector of coefficients to be calibrated from the data. The term $\epsilon$ is a random error with mean zero added for bias.

The model uses regression per weekday and incorporates it with local regression, namely, hourly regression. First of all, the model tries to capture the regression of each weekday with a polynomial of degree 10. The hourly regression of the flow is learned per weekday as well with a polynomial of degree 5. As a result, the predicted values are given by combining both values (daily and hourly) with more emphasis on daily regression to avoid over-fitting. We will be using this model to predict the missing values at a given timestamp, then impute them accordingly.

**4. Data and Reconstructed Data**

The work done in this paper is part of the DiSCO2 project currently conducted at the Center for Industrial Mathematics (ZeTeM) at the University of Bremen. The aim of the project is to model the traffic in the city of Bremen in order to make accurate forecasts of different characteristics of traffic, especially traffic flow. The ultimate goal of the project is to set the stage for decision makers to take actions targeting the reduction of $CO_2$ emissions in the context of fighting against climate change and air pollution.

### 4.1. Data Description

In this project, we have large datasets of around 5 years' worth of data. The data is gathered from over 550 measurement sites all around the city, on each of which an inductive loop detector is installed. This data is mainly delivered by the Traffic Management Center of Bremen (VMZ), which is an associated partner in our project.

Figure 2 displays, in red bullets, the location of loop detectors installed all around Bremen to record traffic attributes. This paper is only concerned with one place of the city located in the city center. We focus on a junction surrounded by seven loop detectors (MS217–MS223), as shown in Figure 3. This junction is situated in front of the main train station as well as tram and bus stations, which makes the traffic in this area very messy and subject to a lot of factors. Traffic lights are highly present in this region, but unfortunately we have no data about them. As in any data gathering device, due to malfunctioning, repairing, or data transmission, many entries are missing in the final output recorded in databases. In our case, an important part of the data is missing over all 5 years. Sometimes values are missing for months, and further, the completeness level of many of the detectors is less than 50%. For this reason we selected a time frame where the data to be used is almost complete (98%). First, we will use this data to train and test the predictive model. Afterward, we will destroy parts of this dataset and then try to reconstruct it with the different imputation techniques mentioned in the previous section. Detectors take measurements each 90s, however in our study we use 10 min accumulations. The precise dates used are from 9 April 2018 to 24 June 2018, which covers a period of 11 weeks. As we already mentioned, this period corresponds to the time frame having the least amount of missing entries. In order to measure the performance of the imputation techniques and the accuracy of the forecasts delivered by the predictive model, we divide our dataset into two parts. The first one consists of 8 weeks used for training, followed by 3 weeks for testing. Precisely, training takes place from 9 April 2018 to 3 June 2018 then we test for the period going from 4 June 2018 to 24 June 2018. The best imputation strategy will be later used to fill in missing, corrupted, and outlier values in our database.

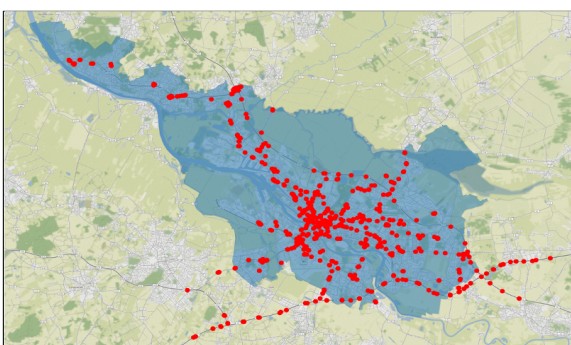

**Figure 2.** Location of the detectors (in red bullets) installed all over Bremen.

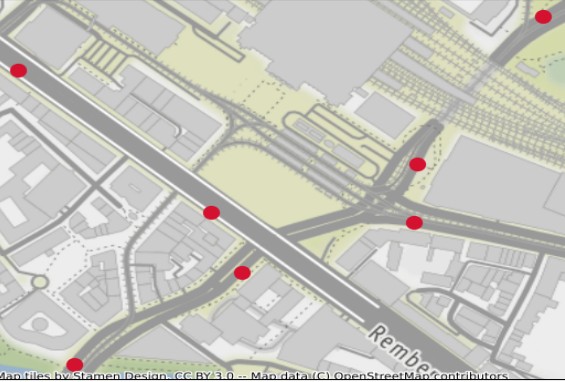

**Figure 3.** Detectors location in the studied junction.

### 4.2. Reconstructed Data

In this subsection, we describe how we destroy parts of the working dataset and reconstruct it. First, from the data described in the previous subsection, we take out the same time frame where we have almost complete data (98%). We then artificially create incomplete datasets by randomly removing different portions of data to reach a certain level of completeness. Since in our data we have different lengths of missing portions, ranging from one timestamp to even months, we will proceed by using analogous reasoning. We proceed by removing, at random, timestamp portions of one of the following sizes, $\{1, 3, 6, 36, 72, 144, 288, 1008, 4320\}$, respectively corresponding to 10 min, 30 min, 1 h, 6 h, 12 h, 1 day, 2 days, 1 week, and 1 month periods of time. The deletions are carried out at random points until we reach different levels of completeness: 50%, 60%, 70%, 80%, and 90%. We also construct incomplete datasets to reach the previous incompleteness ratios by passing in a list of random portion sizes, hence, there are different interval lengths of missing data in each dataset.

Via what we have just described, we create 50 variants of incomplete datasets having different combinations of gap lengths and incompleteness levels. To each of these we apply the imputation techniques reported in Section 3 to construct complete datasets.

### 4.3. Performance of Imputation Methods

In order to measure the performance of the imputation methods, we consider the same split mentioned above for our dataset. The first one is used to train the methods, and the second part is to test their performance. We use mean absolute error (MAE), given in Equation (12) as a criterion of accuracy. From the results reported in Tables 1–3 and their corresponding Figures 4–6, we clearly see that the three methods are closely competitive; however, it is obvious that the linear regression model is more accurate than the others. The performance of the three models varies in function of completeness level and gap lengths as well. Therefore, in what follows we comment and discuss the results based on these attributes.

- Completeness ratio: The experiments reported in Tables 1 and 2, respectively plotted in Figures 5 and 6, used different levels of completeness to investigate the impact of various missing portions of data. The results showed that the completeness percentage has an influence on the accuracy of the imputation methods. As we increase the number of missing entries, the performance quality of the three imputation methods decreases from around 91 with 50% completeness to 51 with 90% completeness. This is clearly apparent in Figure 6, where a list of random gap lengths is passed in. In contrast to that, Figure 5 shows that there is only a slight impact on the completeness ratio when deletions are based on fixed gap lengths. This kind of performance is mainly due to the large gaps of deletions (week and month), in this case deletions sometimes take place mostly in the training set and sometimes in the test set, which alternates the performance quality.
- Gap lengths: The results exhibited in Table 3 and Figure 4 suggest that for small gap deletions the performance of the models is worse than the one with larger gaps. When gap length is between 10 min and 1 day, the MAE is between 75 and 80, however, it drops down to around 72 for one week gap and 58 for one-month deletion. This kind of performance suggests, first, that the deletion of whole consecutive days has a smaller impact on the performance of the models than missing shorter entries for one day. Secondly, this means that training with smaller complete datasets is better than doing it with larger ones with multiple missing entries of a length less than one day. The efficiency of the models gets even better when the gap gets larger, namely one week and one month. In these cases, two possibilities are to be considered. The first is that the deletions are mostly (due to their length: a week or a month) in the training set, which means that only a few entries on the test set have to be imputed, which explains low errors (MAE). The other is that more missing entries are located in the

test set, thus the training set is somehow complete, which affected well the filling process of the missing values in the test set.

**Table 1.** Performance (MAE) of imputation methods in function of completeness ratio on fixed gap-lengths datasets.

| Completeness Ratio | 50% | 60% | 70% | 80% | 90% |
|---|---|---|---|---|---|
| Mean | 77.28 | 78.71 | 76.24 | 79.11 | 79.09 |
| Mean Weekday | 78.26 | 75.80 | 74.04 | 79.11 | 76.46 |
| Linear Regression | **76.95** | **74.82** | **73.26** | **78.42** | **76.01** |

**Table 2.** Performance (MAE) of imputation methods in function of completeness ratio on datasets with a list of gap-lengths.

| Completeness Ratio | 50% | 60% | 70% | 80% | 90% |
|---|---|---|---|---|---|
| Mean | 92.13 | 69.48 | 71.62 | 64.90 | 51.62 |
| Mean Weekday | 91.56 | 57.82 | 66.50 | 61.83 | 51.60 |
| Linear Regression | **91.22** | **57.27** | **66.50** | **61.44** | **51.50** |

**Table 3.** Performance (MAE) of imputation methods in function of gap-length.

| Gap Length | Mean | Mean Weekday | Linear Regression |
|---|---|---|---|
| 1 | 78.50 | 77.89 | **76.85** |
| 3 | 78.62 | 79.70 | **78.33** |
| 6 | **78.33** | 79.42 | 78.40 |
| 36 | 79.90 | 80.47 | **79.65** |
| 72 | 80.45 | 79.98 | **78.85** |
| 144 | 77.49 | 77.03 | **76.39** |
| 288 | 78.13 | 77.16 | **76.56** |
| 1008 | 76.51 | 72.55 | **71.99** |
| 4320 | 72.12 | 58.83 | **58.10** |

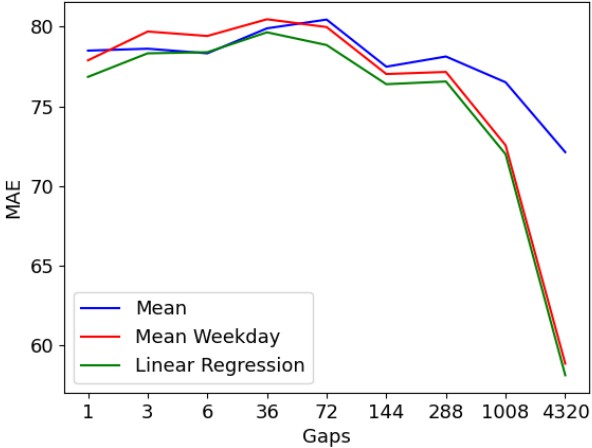

**Figure 4.** Performance of imputation methods in function of gap-length.

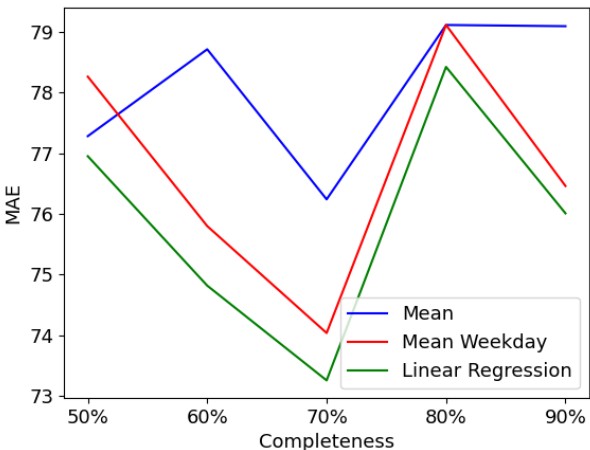

**Figure 5.** Performance of imputation methods for fixed gap-length.

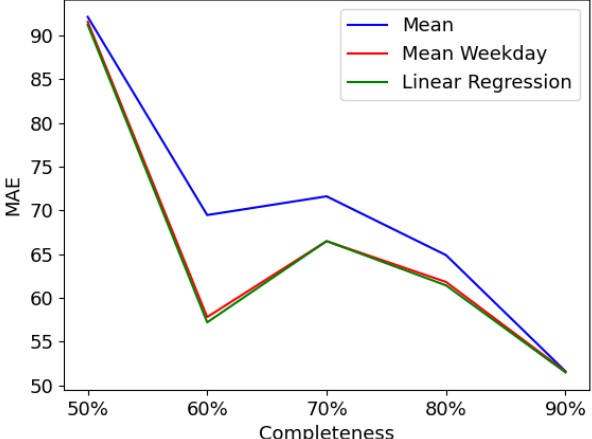

**Figure 6.** Performance of imputation methods for a list of gap-lengths.

### 4.4. Deviation between Original and Reconstructed Data

As introduced above, we artificially produced 50 datasets with multiple kinds of deletions, including fixed gap lengths and a list of random gaps under different percentages of completeness. Afterward, different models were applied to reconstruct missing entries in these datasets. This subsection quickly comments on some significant samples of distributions of original and reconstructed datasets. Distributions are plotted detector-wise, wherein the plots show the deviation between original, incomplete, and reconstructed datasets. The results are given in function of both completeness level and gap length, however, for brevity, herein we only include a few plots.

The plots in Figure 7 are taken from detector MS219 and aggregated by the percentage of completeness. We can clearly see that as we increase the percentage of incompleteness, the deviation between original and reconstructed datasets tends to grow, and vice versa. Although the filling methods are closely competitive, we can notice that linear regression has the least deviation from the original data, accordingly to what has been reported above. Similar conclusions can be drawn as we increase the gap length as well, as shown in Figures 8 and 9 taken from detector MS218. Note that this is also the case for almost all the other detectors.

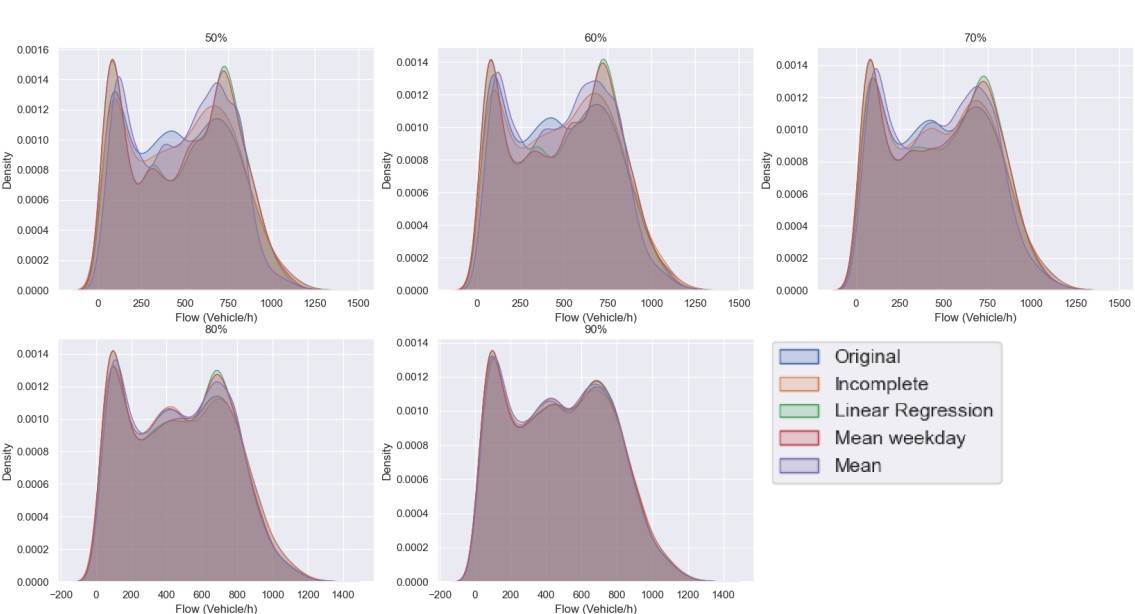

**Figure 7.** Deviation between original and reconstructed datasets with different missing portions: Detector MS219.

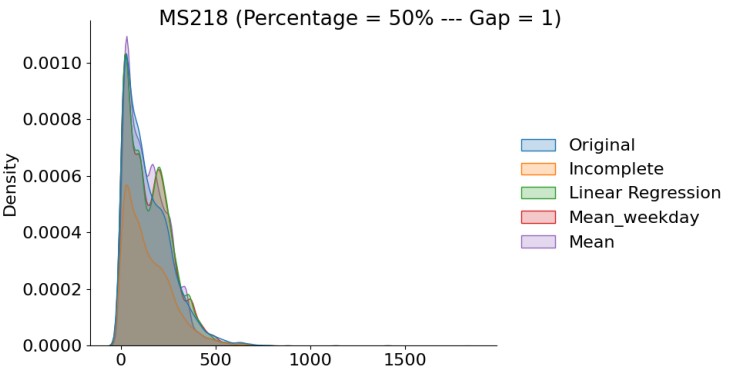

**Figure 8.** Distribution of original and reconstructed data with 50% completeness and gaps of length 1.

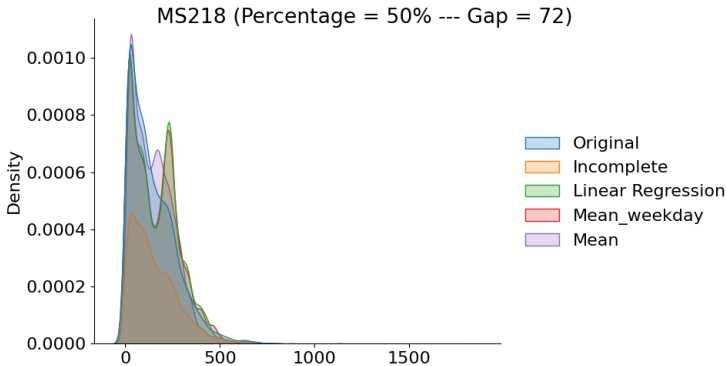

**Figure 9.** Distribution of original and reconstructed data with 50% completeness and gaps of length 72.

In order to give more insights into the deviation between original and reconstructed data, Figures 10–12 exhibit how the linear regression model reconstructs data. The figures are samples taken from the same day (9 April 2018) and different detectors (MS217, MS220,

and MS223). The data has 70% completeness level with missing entries of gap 1, 36, and 72 timestamps. We can notice that when we only have gaps of one timestamp missing at once, the data is somehow well reconstructed. As we increase the gap, the accuracy of the LR technique decreases. Figure 11 shows two gaps of 36 timestamps missing. We can see that the original data is very sparse, however, the LR model tries to reduce the effect of potential noise and outliers by imputing less sparse values. This is also set to avoid over-fitting as shown in Figure 12 as well.

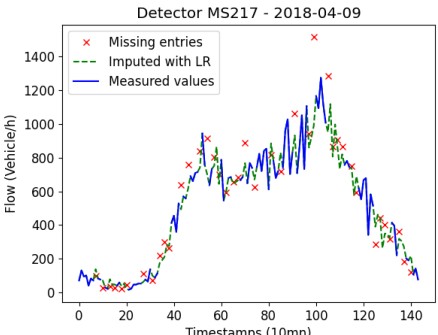

**Figure 10.** Data reconstructed with the Linear Regression method (completeness = 70%, gap = 1).

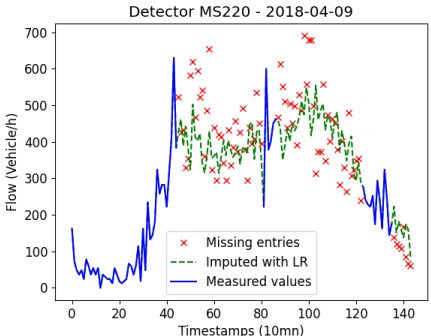

**Figure 11.** Data reconstructed with the Linear Regression method (completeness = 70%, gap = 36).

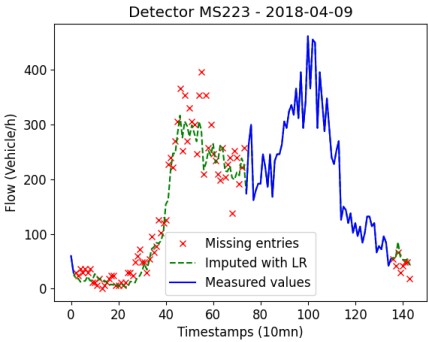

**Figure 12.** Data reconstructed with the Linear Regression method (completeness = 70%, gap = 72).

## 5. Results and Discussion

In this section, we report and discuss the output of our implemented model under both original and artificial datasets. Before doing so, in order to evaluate the accuracy of the predictions, two metrics are deployed: Mean Absolute Percentage Error (MAPE) and Mean Absolute Error (MAE). These are given by the following formulas:

$$MAPE = \frac{100}{n} \cdot \sum_{i=1}^{n} \left| \frac{m(t)_i - p(t)_i}{m(t)_i} \right| \qquad (11)$$

$$MAE = \frac{1}{n} \cdot \sum_{i=1}^{n} |m(t)_i - p(t)_i| \tag{12}$$

Such that $m(t)$ is the real value of traffic flow measured at instant $t$ and $p(t)$ is the value predicted by the model. $n$ is the number of predictions.

*5.1. Under Original Data*

We report and comment in this subsection on the performance of the E-KNN model under the original dataset described in the previous section. We use the same dates for training and testing, namely, we train the model with data from 9 April 2018 to 3 June 2018, then test it on data from 4 June 2018 to 24 June 2018. The discussion of the output of E-KNN will be carried out according to the mean average error (MAE) and mean absolute average error (MAPE) given above (refer to Figures 13 and 14 for more details). We launch the E-KNN model to run on each detector separately because, as can be seen from Figure 1, the data is different from one detector to another. This is due to the location where the detectors are installed. Some loop detectors are collecting data from one-lane roads, whereas others record it from two-lane roads. Thus, the order of magnitude of the flow volume is different for each detector. Predictions are made for one hour, which means we forecast six steps at once. As structured in Table 4, we report the performance accuracy based on day hours. we consider traffic flow for all day hours, significant traffic hours (between 6 h in the morning and 22 h in the evening), morning peak times (from 6 h to 9 h), and finally evening peak times (from 16 h to 19 h). It is obvious that the results vary from one detector to another, and this is related, firstly, to the same reasons just mentioned (distinct flows), and secondly to traffic lights, which we have no data on. Therefore, the accuracy of E-KNN is henceforth discussed by averaging the results in each category. First of all, one can notice that when the flow volume is significant the MAE tends to grow while the MAPE tends to decrease. Since the flow volume in our data is relatively large, it can reach more than 1500 vehicles/hour (see Figure 1), we can expect this kind of somehow large MAEs. Note that the results have also been affected by the occasional outliers, which we did not remove from our dataset. Taking into account all day hours, the E-KNN model makes forecasts with an error of 46.40 MAE and 26.38% MAPE. However, when we only regard significant traffic during the day, meaning traffic between 06:00 and 22:00, a worse MAE is reached (56.26) and a better MAPE is delivered (17.91%). It is expected to have a worse MAE because during this period of time larger numbers of vehicles are flowing. The accuracy in this category of hours is satisfactory as it reaches around 83%. Moreover, two other categories appear as rush hours during the day, in particular, peak times in the morning and in the evening, from 06:00 to 09:00 and from 16:00 to 19:00. Relative absolute error in the morning is around 22%, whereas it is only 16% in the evening. Given that the detectors are installed on signalized urban arterial roads and we have no data on how traffic lights are programmed, we think that the accuracy of the E-KNN model is satisfactory, especially during rush hours.

**Table 4.** Prediction results over the test set for 1 h (6 steps).

| Detector ID | 06:00–09:00 | | 16:00–19:00 | | 06:00–22:00 | | All Day | |
|---|---|---|---|---|---|---|---|---|
| | MAE | MAPE | MAE | MAPE | MAE | MAPE | MAE | MAPE |
| MS217 | 63.58 | 14.30 | 100.50 | 12.10 | 75.08 | 11.97 | 62.87 | 16.64 |
| MS218 | 23.96 | 28.86 | 55.94 | 20.29 | 36.11 | 22.78 | 28.80 | 30.57 |
| MS219 | 61.54 | 15.34 | 92.90 | 11.98 | 71.28 | 12.35 | 60.41 | 18.80 |
| MS220 | 73.98 | 21.12 | 80.82 | 17.16 | 74.10 | 18.30 | 62.32 | 27.12 |
| MS221 | 26.24 | 25.25 | 53.18 | 16.99 | 45.87 | 21.58 | 38.24 | 31.30 |
| MS222 | 38.42 | 26.63 | 57.00 | 17.77 | 47.59 | 19.31 | 37.11 | 30.54 |
| MS223 | 47.54 | 23.98 | 49.78 | 17.45 | 43.80 | 19.09 | 35.06 | 29.68 |
| Average | 47.89 | 22.21 | 70.02 | 16.25 | 56.26 | 17.91 | 46.40 | 26.38 |

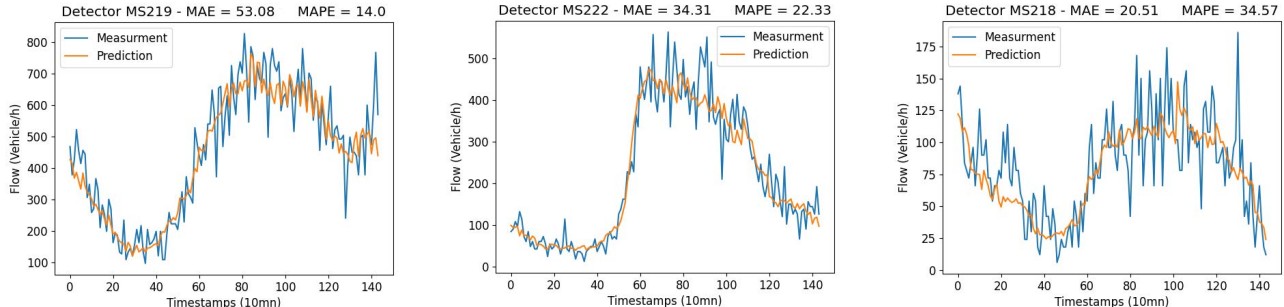

**Figure 13.** Predictions of traffic flow on randomly chosen detectors (weekend days).

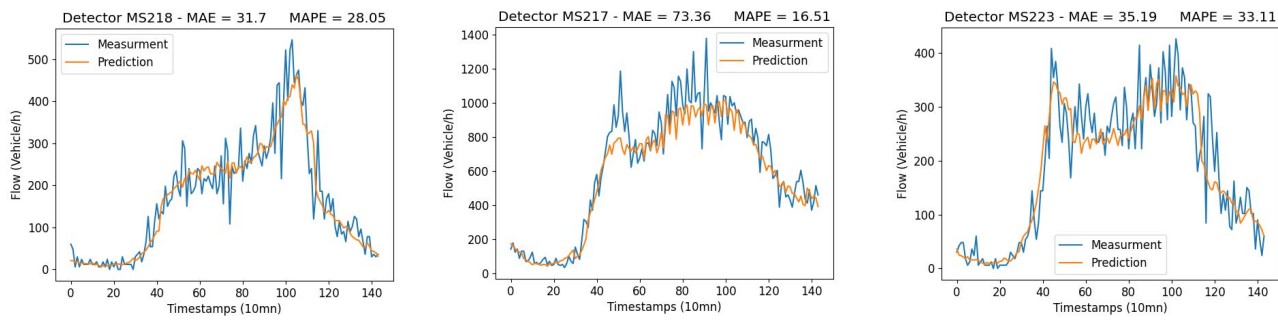

**Figure 14.** Predictions of traffic flow on randomly chosen detectors (working days).

### 5.2. Under Artificial Datasets

This subsection is dedicated to the discussion of E-KNN performance under the different imputed datasets produced as described in Section 4.2. In order to assess the performance of E-KNN under the filled-in datasets (50 datasets), we apply it to each of these datasets, always using the same time frames declared above for training and testing. The accuracy of the model is reported using the mean absolute error criterion (MAE) since both MAE and MAPE gave equivalent output. The results are discussed according to the level of completeness of each dataset and also following gap lengths. The results show that the ratio of completeness has an impact on the performance of E-KNN, however, not for all of the 50 datasets. This is also related to the positions of deletions (especially for large gaps) where sometimes they lay mostly in the training set and other times in the test set. Another factor that should be taken into account is the flow volume, which varies from one detector to another (refer to Figure 1) and sometimes leads to differences in errors when deletions take place mostly in either the training set or test set. From Tables 5 and 6, in general, as we increase the level of completeness of the datasets, E-KNN seems to perform better. The model's performance also depends on the length of gaps; for some of these configurations, the completeness level seems to have a huge impact on the performance ($gap = 36$, $gap = 1008$), and a slight one on others, as in $gap = 1$ for instance (see Figure 15). We think that this does not relate to the gap length itself but to the random distribution of deletions between the test set and training set in these datasets. Apart from that, completeness level has a huge impact, especially on incomplete datasets. For instance, on incomplete datasets with $gap = 144$, MAE decreases from 200 on a 60% level to around 80 on 90% completeness. Moreover, on datasets completed with the three models, MAE decreases from around 60 to 50 as we increase the completeness ratio. The same thing can be noticed with $gap = 36$ (see Figure 15), where for incomplete datasets MAE decreases from 60 to 55 and from around 65 to 47 for completed datasets. For $gap = 1008$ (see Figure 15), completeness level seems to have no effect on incomplete datasets, but intensely impacts the performance of E-KNN with filled-in datasets as MAE decreases from 65 to around 47.

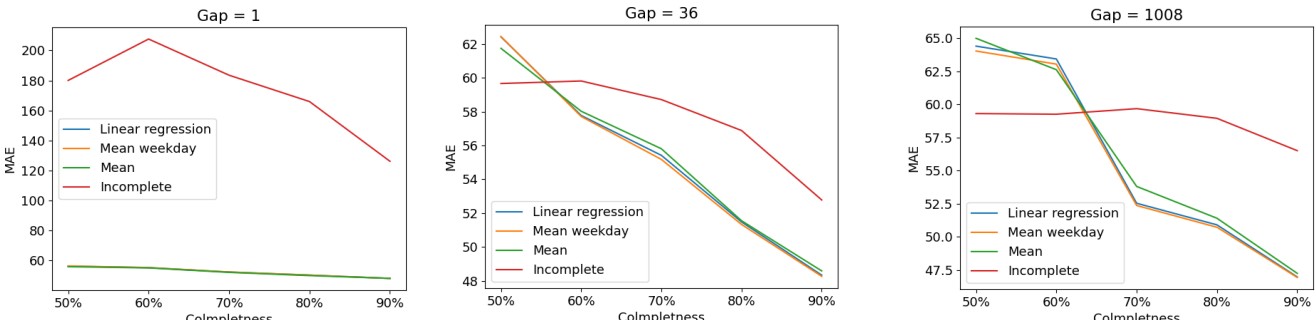

**Figure 15.** Performance of E-KNN under incomplete and completed datasets with different gap-lengths.

On the other side, the gap length parameter also has an impact on E-KNN performance. In general, when gap length is increased, the accuracy decreases (refer to Table 7). However, this is not the case for our results sorted in function of completeness level. It seems that, in presence of filled-in datasets, the gap length has little impact on the performance of E-KNN which means that the datasets are somehow efficiently completed regardless of the different deletion kinds. Howbeit, for incomplete datasets, gap length has an effect on the accuracy. Take for instance 90% completeness (refer to Figure 16), MAE increases as we increase the gap length, except for 72 and 288 where training sets seem to have fewer deletions than test sets, which allowed E-KNN to perform well. In this case, MAE jumps from around 50 on gap 1 to around 200 on gap 4320.

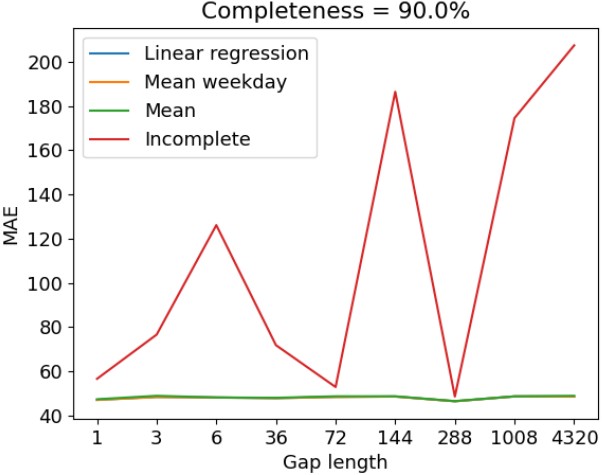

**Figure 16.** Performance of E-KNN on incomplete and completed datasets with a completeness level of 90%.

**Table 5.** E-KNN's performance for imputation methods in function of completeness ratio on datasets with a list of gap-lengths.

| Completeness Ratio | 50% | | 60% | | 70% | | 80% | | 90% | |
|---|---|---|---|---|---|---|---|---|---|---|
| | MAE | MAPE | MAE | MAPE | MAE | MAPE | MAE | MAPE | MAE | MAPE |
| Incomplete | 157.19 | 158.73 | 150.03 | 127.85 | 102.47 | 114.57 | 120.49 | 112.31 | 58.73 | 37.23 |
| Mean | **53.56** | 32.32 | 54.09 | 34.57 | 53.10 | 33.00 | 51.16 | 30.14 | 46.97 | 27.93 |
| Mean Weekday | 53.76 | 29.24 | **51.65** | 29.45 | **52.12** | 29.81 | **50.29** | 27.39 | **46.69** | 26.35 |
| Linear Regression | 53.79 | **29.10** | 51.72 | **29.38** | 52.25 | **29.72** | 50.46 | **27.21** | 46.73 | **26.30** |

**Table 6.** E-KNN's performance for imputation methods in function of completeness ratio on datasets with fixed gap-lengths.

| Completeness Ratio | 50% | | 60% | | 70% | | 80% | | 90% | |
|---|---|---|---|---|---|---|---|---|---|---|
| | MAE | MAPE | MAE | MAPE | MAE | MAPE | MAE | MAPE | MAE | MAPE |
| Incomplete | 124.91 | 111.74 | 167.87 | 139.26 | 159.17 | 159.63 | 142.12 | 131.63 | 111.20 | 93.35 |
| Mean | 60.35 | 38.45 | 56.99 | 35.66 | 53.69 | 33.49 | 50.79 | 30.55 | 48.09 | 28.01 |
| Mean Weekday | 60.15 | 33.34 | **56.58** | 31.61 | **53.18** | 29.95 | 50.63 | 28.40 | **47.87** | 26.83 |
| Linear Regression | **60.09** | **33.17** | 56.62 | **31.48** | 53.28 | **29.85** | 50.27 | **28.34** | 47.91 | **26.78** |

**Table 7.** E-KNN's performance in function of gap length.

| Gap Length | Incomplete | | Mean | | Mean Weekday | | Linear Regression | |
|---|---|---|---|---|---|---|---|---|
| | MAE | MAPE | MAE | MAPE | MAE | MAPE | MAE | MAPE |
| 1 | 58.73 | 32.74 | **52.16** | 32.03 | 52.41 | 29.71 | 52.36 | **29.58** |
| 3 | 57.57 | 32.29 | **53.22** | 32.75 | 53.74 | 30.30 | 53.62 | **30.16** |
| 6 | 174.61 | 207.74 | **53.34** | 32.62 | 53.92 | 30.41 | 53.78 | **30.27** |
| 36 | 235.41 | 179.15 | 55.14 | 34.36 | **54.98** | 31.06 | 55.08 | **30.98** |
| 72 | 219.87 | 215.39 | 55.57 | 34.62 | **54.89** | 31.07 | 55.02 | **31.01** |
| 144 | 172.62 | 166.38 | 55.12 | 33.47 | **54.71** | 30.01 | 54.84 | **29.95** |
| 288 | 151.30 | 157.05 | 55.37 | 33.92 | **54.11** | 29.60 | 54.24 | **29.51** |
| 1008 | 144.13 | 139.71 | 56.00 | 35.25 | **55.41** | 30.79 | 55.64 | **30.74** |
| 4320 | 70.35 | 47.85 | 49.92 | 30.08 | 48.97 | 27.27 | **48.95** | **27.12** |

In a nutshell, based on the discussion above, we can say that the E-KNN model is very sensitive to incomplete datasets. This sensitivity also varies based on both the completeness level of datasets and gap sizes. In general, as we increase completeness level, the accuracy increases, and vice versa. The same thing might apply to gap lengths, as we increase them, accuracy tends to decrease; however, not if the datasets are well reconstructed. For completed datasets, E-KNN performance is often somehow stable under different missing percentages and gap lengths; however, sometimes these two parameters also impact E-KNN's performance in a similar manner as happens with incomplete datasets. When the performance of E-KNN is stable with regard to different gaps and completeness ratios, it means that the imputation techniques are efficient and well reconstructed the datasets despite the different factors considered.

## 6. Conclusions

We investigated in this paper the impact of data loss on the performance of the K-nearest neighbors model applied to the context of intelligent transportation. The model delivers a multi-step flow forecast for urban roads located in downtown Bremen. In order to examine the efficiency of our E-KNN model under data loss circumstances, we artificially created incomplete datasets with different completeness levels and gap lengths. Afterward, we designed three different methods for the sake of reconstructing these datasets. The performance of the E-KNN model is then tested with the original, incomplete, and imputed datasets. The experimental results showed that E-KNN is able to reach 17% MAPE during significant daily traffic (06:00–22:00) for 3 weeks of the test set. Moreover, considering all day hours, the model is able to make 6-step forecasts with an average error of one car per 90 s. The performance of the model under imputed datasets varies in function of completeness level and gap length, but in general the model performs much better under filled-in datasets than under incomplete ones.

**Author Contributions:** Conceptualization, A.M. and D.K.; Funding acquisition, C.B.; Methodology, A.M. and D.K.; Project administration, C.B.; Supervision, C.B.; Writing—original draft, A.M.; Writing—review & editing, A.M. and D.K. All authors have read and agreed to the published version of the manuscript.

**Funding:** DiSCO2 project is funded by the European Regional Development Fund (ERDF).

**Institutional Review Board Statement:** Not applicable.

**Informed Consent Statement:** Not applicable.

**Data Availability Statement:** Not applicable.

**Conflicts of Interest:** The authors declare no conflict of interest.

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
