# Peer review of "Impact of Data Loss on Multi-Step Forecast of Traffic Flow in Urban Roads Using K-Nearest Neighbors"

_sustainability, doi:10.3390/su141811232_

Round 1

Reviewer 1 Report

This paper examines the impact the impact of data loss on the behavior of one of the frequently used approaches, namely, k-Nearest Neighbors (KNN) model. To make KNN more adaptive to traffic flow prediction problem, this paper proposes an enhanced k-Nearest Neighbors model which incorporate some characteristics. Also, some imputation techniques are introduced by authors. Comprehensive experiments are conducted, which show the impact of data loss and the performance of proposed model.

However, there are no comparation experiments between different models which lacks integrity in terms of the performance of propose model. Moreover, the figures are not informative and some metrics like variance should be incorporated.

Overall, the paper is well written. The reviewers consider it can be published in Sustainability after some revisions to above questions.

Reviewer 2 Report

A well-written paper, presenting the results of a fresh approach to transport analysis. It can be published with minor modifications only: a) the legends with the five categories in figures 7-9 need to be enlarged and b) a discussion section should be added.
